

# Tensor network variational optimizations for real-time dynamics: Application to the time-evolution of spin liquids

**Ravi Teja Ponnaganti, Matthieu Mambrini and Didier Poilblanc***

Laboratoire de Physique Théorique UMR5152, C.N.R.S. and Université de Toulouse, 118 rte de Narbonne, 31062 Toulouse, France

⋆ didier.poilblanc@irsamc.ups-tlse.fr

## Abstract

Within the Projected Entangled Pair State (PEPS) tensor network formalism, a simple update (SU) method has been used to investigate the time evolution of a two-dimensional U(1) critical spin-1/2 spin liquid under Hamiltonian quench [Phys. Rev. B 106, 195132 (2022)]. Here we introduce two different variational frameworks to describe the time dynamics of SU(2)-symmetric translationally-invariant PEPS, aiming to improve the accuracy. In one approach, after using a Trotter-Suzuki decomposition of the time evolution operator in term of two-site elementary gates, one considers a single bond embedded in an environment approximated by a Corner Transfer Matrix Renormalization Group (CTMRG). A variational update of the two tensors on the bond is performed under the application of the elementary gate and then, after symmetrization of the site tensors, the environment is updated. In the second approach, a cluster optimization is performed on a finite (periodic) cluster, maximizing the overlap of the exact time-evolved state with a symmetric finite-size PEPS ansatz. Observables are then computed on the infinite lattice contracting the infinite-PEPS (iPEPS) by CTMRG. We show that the variational schemes outperform the SU method and remain accurate over a significant time interval before hitting the entanglement barrier. Studying the spectrum of the transfer matrix, we find that the asymptotic correlations are very well preserved under time evolution, including the critical nature of the singlet correlations, as expected from the Lieb-Robinson (LR) bound theorem. Consistently, the system (asymptotic) boundary is found to be described by the same Conformal Field Theory of central charge $c = 1$ during time evolution. We also compute the time-evolution of the short distance spin-spin correlations and estimate the LR velocity.



# 1 Introduction

The search for spin liquids in condensed matter materials is a very rapidly developing area of quantum magnetism [1]. In a classical magnetic system, the magnetic moments of the constituent particles align to form a well-defined pattern or order, such as ferromagnetism or antiferromagnetism. In contrast, in a spin liquid, the magnetic moments are entangled and do not exhibit any long-range order, despite being at low temperatures. This gives rise to a state of matter that is neither a solid, liquid, nor gas, but rather a "quantum spin liquid." The entangled magnetic moments in a spin liquid are "frustrated," meaning that they are unable to achieve their lowest energy state due to the geometry of the system. This leads to a highly degenerate manifold of many possible configurations of the magnetic moments as in the prototypical Resonating Valence Bond (RVB) state proposed by Anderson [2] which shows no symmetry breaking, even down to zero temperature, due to enhanced zero-point quantum fluctuations. The entangled nature of the spin liquids leads to unique exotic properties [3–5] such as fractionalization of excitations, emergent gauge fields, topological properties, etc. Spin liquids hold promise for applications in quantum computing and information processing [1, 6, 7]. However, they are also challenging to study experimentally due to their elusive and entangled nature [8, 9].

A quantum quench refers to a sudden change in the parameters of the quantum system, leading to a rapid and non-equilibrium evolution of the system. Quantum quenches have been studied extensively in recent years [10–15], both theoretically and experimentally, because they provide a way to probe the non-equilibrium dynamics of quantum systems. They are also relevant to a wide range of physical systems, including condensed matter systems,

ultracold atomic gases, and quantum field theories. Nowdays, quantum simulators based on cold atoms on two-dimensional (2D) optical lattices are being realized experimentally with the realistic perspective of emulating simple models of condensed matter physics [16–20]. Rydberg atoms platforms are also used to realize dimer liquids or spin liquids [21, 22]. It is therefore necessary to develop new theoretical tools to compute faithfully the time evolution of 2D spin systems in order to address e.g. adiabatic evolutions, quench or Floquet dynamics in various experimental set-ups. In the following we shall address the quantum dynamics of a RVB spin liquid after a quantum quench. Here the quench will be implemented by a sudden change in the Hamiltonian that governs the system's behavior, leading to a rapid change in the system state. Then, the system is driven out of its equilibrium state, and the time evolution of the system becomes very complex and difficult to predict.

The variational methods we describe here apply to simple quench set-ups starting from an initial quantum state $|\Psi_0\rangle$ preserving lattice and SU(2)-rotation symmetries, namely a spin liquid. Here, as a simple example, we shall consider a nearest neighbor (NN) Resonating Valence Bond (RVB) spin liquid on an infinite square lattice. The NN RVB state consists of resonating NN singlets and is a special point of an enlarged spin liquid family [4] including longer-range singlet bonds as well. When only NN bonds are present, the RVB state shows long-range dimer correlations originating from a local U(1) gauge symmetry. Recent work [4, 23] suggested that topological order appears immediately whenever longer-range singlets are present breaking the U(1) gauge symmetry to $\mathbb{Z}_2$.

Our methods deal with quench Hamiltonians which preserve both lattice and spin symmetries. Such symmetries in the initial state and in the quench Hamiltonian will be used explicitly in the procedure. However, the preservation of the U(1) gauge symmetry depends on the method used as discussed later. To illustrate the methods we shall consider a simple quench protocol,

$$\mathcal{H}(t) = \begin{cases} 0, & \text{for } t \leq 0, \\ H = J \sum_{\langle x,y \rangle} \mathbf{S}_x \cdot \mathbf{S}_y, & \text{for } t > 0, \end{cases} \tag{1}$$

where $H$ is the NN Heisenberg model on the square lattice, and take advantage of the small entanglement (bounded by $\ln 3$ per bond) of our initial state as well as the full lattice and spin symmetries to study the time evolution $|\Psi(t)\rangle = \exp(-iHt)|\Psi_0\rangle$ over a small time $t > 0$ interval. Note that, hereafter, the time $t$ will be expressed in units of the inverse-coupling $1/J$.

In recent years, progress have been made in developing 2D tensor network methods for real and imaginary time evolution. In particular, Projected Entangled Pair States (PEPS) on the infinite lattice (infinite-PEPS or iPEPS) have been used to study the dynamics in the 2D quantum Ising model after a quench from a fully polarized (product) state [24, 25], with the goal of (approximately) maximizing the overlap of the PEPS with the exact time-evolved state. Very recently, in order to go beyond the previous simplified schemes, the optimization was performed in a tangent space of the iPEPS variational manifold [26]. Our current developments follow the same conceptual ideas but take into account explicitly all the symmetries in the problem to improve the accuracy and efficiency of the variational optimization scheme. Note also that our initial state is a correlated entangled state, in contrast to most studies starting from a product state. The new schemes will be compared to the simplest Simple Update (SU) scheme, which will serve later on as a reference.

## 2 Methods

### 2.1 Symmetric PEPS Ansatz and summary of SU results

In a recent paper [27] we used the SU method to investigate the time evolution of the NN RVB state under the Hamiltonian quench defined in Eq. (1) using symmetric PEPS. The method is based on: (i) a classification of SU(2) invariant and $C_{4v}$-symmetric tensors [28], (ii) the identification of a tensor manifold (PEPS ansatz) shown to be relevant to capture the quench dynamics at small times, (iii) a SU procedure allowing to compute the time evolution within the PEPS manifold defined by the site tensor class. In this section, we briefly recall these three steps. For a more complete description the reader may refer to reference [27].

*Symmetric tensors classification -* Both the initial state and the Hamiltonian governing the dynamics of our problem are SU(2) symmetric and transform according to the trivial representation of the square lattice point group $C_{4v}$. It is therefore crucial to enforce these key properties at every step of our scheme. The trivial representation of $C_{4v}$ is simply obtained by choosing a uniform site tensor ansatz on the lattice. As explained in details in Ref. [28], the continuous SU(2) of PEPS can be implemented a the level of local tensors by imposing that the $D$-dimensional subspace of each virtual legs has the structure of a reducible representation of SU(2), namely a direct sum of SU(2) irreducible representations. As an example, the NN RVB state has a simple representation using $v = 0 \oplus 1/2$ virtual bonds ($D = 3$). More generally for any given $v$, it is possible to classify tensors and define manifolds in which all tensors are linearly independent and mutually orthogonal. Working in a given manifold not only allows to fix the PEPS symmetry but also greatly reduces the number of parameters describing the PEPS family. This last point is a major advantage for computations based on optimization.

*Local tensor ansatz and SU method -* As explained in Ref. [27] and recalled in subsection 2.2.1, the time evolution unitary operator is split using a Trotter-Suzuki (TS) decomposition in a product of 2-site unitary gates. Once applied on two neighboring tensors, the virtual bond dimension is no longer $D = 3$ but still has the structure of a SU(2) reducible representation. In order to elucidate the SU(2) content of this updated virtual bond, this 2-site object can be reinterpreted as a *symmetric complex* matrix that, in turn, has to be reduced. To keep a uniform PEPS representation (i.e. with same tensor on every lattice site), a symmetric reduction is required. This leads to a technical difficulty: the non-Hermiticity of the matrix prevents diagonalization, and on the other hand conventional SVD leads to a non-uniform representation. This can be circumvented by using a symmetric SVD (Autonne-Takagi decomposition [29–31]). For small TS time step, the analysis reveals that the relevant virtual subspace [27] is $0 \oplus 1/2 \oplus 1$ which correspond to $D = 6$. This subspace defines a $M = 11$ dimensional $C_{4v}$-symmetric tensor manifold that generalizes and includes the initial $D = 3$ tensor manifold. Due to SU(2) fusion rules the number of half-integer spins hosted on the four virtual bonds has to be odd. In this case, only tensors with one or three $1/2$ spins are allowed. This defines a $\mathbb{Z}_2$ symmetry associated to the gauge symmetry operator $Z = \prod_{i=1}^{4} Z_i$ acting on the four virtual spaces of the site tensor, where $Z_i$ takes the form $Z_i = (-1)^{n_i}$, $n_i$ being the number of spin-$1/2$.

Interestingly, the dynamics computed in the SU scheme [27] selects a $\tilde{M} = 8$ tensors submanifold corresponding to a U(1) gauge symmetry where the total number of $1/2$ spins hosted on the four virtual bonds is conserved and fixed to one. This fact can be understood from a Projected Entangled Pair Operator (PEPO) representation of the 2-site Heisenberg gate whose virtual bond ($D = 4$) has the structure $0 \oplus 1$ involving only integer spins. Such a gate cannot change the integer (resp. half-integer) spin nature of the updated bond, and hence conserves the number of integer (resp. half-integer) spins hosted by the site tensor virtual bonds.

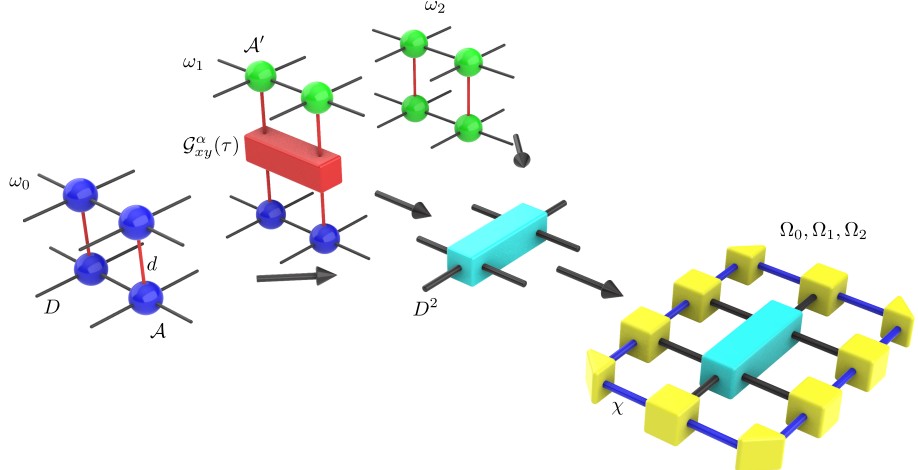

Figure 1: Embedded-bond variational optimization (EBVO): the overlaps $\Omega_\alpha$, $\alpha = 0, 1, 2$, are obtained by embedding the 2-site double layer tensors $\omega_\alpha$ (generically represented as a light blue parallelogram) in a fixed-point environment (in yellow) obtained by CTMRG. $\omega_1$ includes the unitary (two-site) gate.

## 2.2 Embedded-bond variational optimization

### 2.2.1 Trotter Suzuki decomposition

In the embedded-bond variational optimization (EBVO) scheme we start, as for the simple update (SU) scheme, from a usual Trotter-Suzuki (TS) decomposition [32, 33] of the time evolution operator $\exp(-iHt)$ in term of elementary gates

$$\exp(-iHt) = \prod_1^{N_\tau} \exp(-iH\tau) \tag{2}$$

$$\simeq \prod_1^{N_\tau} \mathcal{G}^D(\tau)\mathcal{G}^C(\tau)\mathcal{G}^B(\tau)\mathcal{G}^A(\tau),$$

where $\tau = t/N_\tau$ is a small time step ($\tau \ll 1$) and the Heisenberg Hamiltonian has been split into four parts, $H = H^A + H^B + H^C + H^D$, each acting on one of the four staggered configurations of *disconnected* horizontal or vertical bonds labelled by $\alpha = A, B, C, D$. Note that the second identity involves the standard systematic TS error vanishing in the limit $\tau \to 0$. The four unitaries $\mathcal{G}^\alpha(\tau)$ can then be naturally decomposed in terms of commuting two-site gates acting on nearest-neighbor (NN) $\langle x, y \rangle$ bonds,

$$\mathcal{G}^\alpha(\tau) = \exp(-iH^\alpha\tau)$$
$$= \prod_{\langle x, y \rangle \in C_\alpha} \mathcal{G}^\alpha_{xy}(\tau), \tag{3}$$

where $\mathcal{G}^\alpha_{xy}(\tau) = \exp(-iH^\alpha_{xy}\tau)$. In the current method, at every time step $t$, we focus on a particular $\langle x, y \rangle$ bond (e.g. on the $A$ staggered bond configuration) and update the coefficients $\mu_a(t) \to \mu_a(t + \tau)$ of the tensors on sites $x$ and $y$ under the action of $\mathcal{G}^A_{xy}(\tau)$ defining two new tensors $\mathcal{A}'$ and $\mathcal{A}''$ on sites $x$ and $y$, respectively, related by $180^o$ rotation. To do so one explicitly takes into account the environment of the infinite lattice around the active bond introducing some non-locality (in contrast to SU) using the optimization algorithm described below.

### 2.2.2 Bond optimization

To update the $\mathcal{A}$ tensor one define fidelities i.e. overlaps between ket and bra states as $\Omega_0 = \langle \Psi_{\mathcal{A}} | \Psi_{\mathcal{A}} \rangle$, $\Omega_1 = \langle \Psi_{\mathcal{A}'} | \mathcal{G}_{xy}^A(\tau) | \Psi_{\mathcal{A}} \rangle$ and $\Omega_2 = \langle \Psi_{\mathcal{A}'} | \Psi_{\mathcal{A}'} \rangle$ depicted in Fig. 1. Here the two $\mathcal{A}'$ tensors on sites $x$ and $y$ are SU(2)-symmetric tensors exhibiting mirror symmetry w.r.t. the $xy$ axis ($C_s \subset C_{4v}$ point group) and related by inversion symmetry w.r.t the bond center. Outside of the active 2-site region, all fidelities involve the same uniform tensor network of on-site $C_{4v}$-symmetric double-layer $\mathcal{A}^\dagger \mathcal{A}$ tensor contracted over physical degrees of freedom. This approximation is very well justified when the tensor $\mathcal{A}'$ is close to the exact solution realizing the two site evolution. We have used a single-site (symmetric) Corner Transfer Matrix Renormalization Group (CTMRG) [34–37], more specifically its single-site symmetric version [28], to contract the network around the active bond, resulting into a converged (so-called "fixed-point") SU(2)-symmetric environment of adjustable bond dimension $\chi = \chi_{\text{opt}}$. Note however that the corner transfer matrix $\mathcal{C}_{\text{TM}}$ is here a (non-Hermitian) complex symmetric matrix so that, instead of the usual SVD decomposition, an orthogonal factorization is used,

$$\mathcal{C}_{\text{TM}} = OWO^T, \tag{4}$$

where $O$ is a (non-unitary) complex orthogonal matrix ($OO^T = \text{I}_d$) and $W$ is a complex (eigenvalue) diagonal matrix (see details in Ref [27]). Interestingly, the symmetric character of the transfer matrix is preserved under truncation $\mathcal{C}_{\text{trunc}} = OW_{\text{trunc}}O^T$, keeping in $W_{\text{trunc}}$ the $\chi$ largest (in modulus) eigenvalues of $W$.

Using a conjugate gradient method, the new $C_s$-symmetric tensor $\mathcal{A}'$ is obtained by maximizing (following the lines of the variational optimization scheme [38, 39]) the normalized fidelity $\omega = \Omega_1/(\Omega_0 \Omega_2)^{1/2}$ over the parameters $\{\mu_a'\}$ defining its expansion $\mathcal{A}' = \sum_{a=1}^P \mu_a' T_a'$ in term of the $P \simeq 4M$ elements of the $C_s$-symmetric tensor basis $\{T_a'\}$.

### 2.2.3 Symmetrization

Since the NN bonds of configuration $A$ are disconnected, the same increment of the tensors can be performed on all the A bonds simultaneously. The action of the 3 remaining set of gates $\mathcal{G}_{xz}^B(\tau)$, $\mathcal{G}_{xu}^C(\tau)$ and $\mathcal{G}_{xv}^D(\tau)$ leads approximately to the same increment of the tensor $\mathcal{A}$ at site $x$ along the $xz$, $xu$ and $xv$ bonds obtained by 90, 180 and 270-degrees rotations of the original $xy$ bond with respect to $x$, respectively. More precisely, we can write for the $\mathcal{A}'$ tensor on site $x$:

$$\mathcal{A}' = \mathcal{A}'_\parallel + \delta \mathcal{A}, \qquad \text{where} \qquad \mathcal{A}'_\parallel(t) = \frac{(\mathcal{A}|\mathcal{A}')}{(\mathcal{A}|\mathcal{A})} \mathcal{A}(t), \tag{5}$$

and $(\mathcal{A}|\mathcal{B})$ is the scalar product of $\mathcal{A}$ and $\mathcal{B}$ tensors (contracting over both physical and virtual links). The updated $C_{4v}$-symmetric on-site tensor becomes then

$$\mathcal{A}(t+\tau) \simeq \mathcal{A}'_\parallel(t) + \sum_0^3 R^n(\delta\mathcal{A}), \tag{6}$$

where $R$ is the 90-degree rotation of the local tensor (and $R^0 = Id$). The same procedure applied to the $\mathcal{A}''$ tensor on site $y$ leads to the same tensor so that the updated PEPS remains uniform and symmetric. Note that, to compute observables with the optimized site tensor, a different CTMRG environment can be used with a larger bond dimension $\chi$.

Before moving further to the second variational optimization scheme, it is instructive to discuss the possible sources of error in this method. i) As all other methods discussed here, the fundamental limitation is the truncation of the PEPS ansatz to $D = 6$ which does not allow to accommodate the rapid growth of entanglement of the wavefunction beyond some typical time; one would then hit the "entanglement barrier". ii) Another source of error is

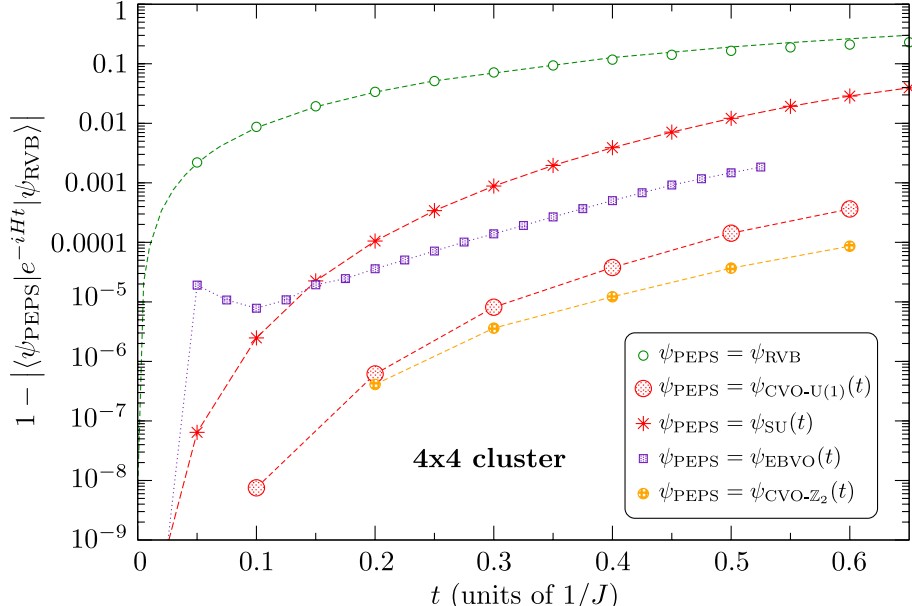

Figure 2: Infidelity (i.e. deviation from 1 of the overlap with the exact time-evolved state) in log scale of various finite-PEPS on a $4 \times 4$ cluster, as a function of time $t$. The reference $\mathcal{I}_0(t)$ (green open circles) is fitted using $\exp(-Ct^{-2})$ (dashed green curve). The Cluster VO has been performed within the U(1) and the enlarged $\mathbb{Z}_2$ PEPS manifolds (filled circles). The EBVO finite-PEPS has been obtained using the U(1) site tensor optimized on the infinite lattice with $\chi = D^2$ and $\tau = 0.025$.

the optimization procedure of the local tensor which, although variational, is done locally. In other words, despite the fact that information about the whole system is included via the environment, the EBVO is intrinsically a local update, the optimization being carried out at *fixed* environment. iii) Lastly, a significant source of error may also come from the simplified symmetrization procedure (5).

## 2.3 Cluster variational optimization

### 2.3.1 Optimization

We now move to the Cluster Variational Optimization (Cluster VO or CVO) algorithm. The idea is to consider a finite periodic cluster and optimize the local tensor of a uniform PEPS defined on that cluster to get the largest fidelity with the time-evolved state. In fact, for a small size cluster like a 16-site $4 \times 4$ torus, the time-evolved state $|\Psi(t)\rangle$, $t > 0$, can be obtained exactly (i.e. up to machine precision) by a series expansion $\sum_{n=0}^{n_{\max}} \frac{(it)^n}{n!} |\Psi_n\rangle$, where $|\Psi_n\rangle = H^n |\Psi_0\rangle$, which converges rapidly as a function of $n_{\max}$. Note that, since the time-evolved state is a spin singlet, the calculation can be performed in the reduced $S_Z = 0$ sector. Note also that the calculation is made simple using the recurrence relation $|\Psi_n\rangle = H|\Psi_{n-1}\rangle$ (i.e. avoiding to compute the operators $H^n$).

To compute the overlap $\Omega_{\mathcal{A}}(t) = |\langle \Psi(t) | \Psi_{\text{PEPS}}(\mathcal{A}) \rangle|$ one needs to first compute the PEPS $|\Psi_{\text{PEPS}}(\mathcal{A})\rangle$ expressed in the $S_Z = 0$ basis (of dimension 12 870 for the 16-site cluster). The optimization of $\Omega_{\mathcal{A}}(t)$ with respect to the coefficients of $\mathcal{A}$ is done using a conjugate-gradient method which requires to recompute $|\Psi_{\text{PEPS}}(\mathcal{A})\rangle$ and $|\Psi_{\text{PEPS}}(\mathcal{A} + \partial \mathcal{A})\rangle$ (to get the numerical gradient of the overlap) along the minimization path in the multi-dimensional parameter space. From this procedure, one eventually gets the optimized tensor $\mathcal{A}^*(t)$ and hence the corresponding optimized PEPS $|\Psi_{\text{PEPS}}(\mathcal{A}^*)\rangle$.

Fig. 2 shows the infidelity $\mathcal{I}(t) = 1 - \Omega_{\mathcal{A}}(t)$ defined by the deviation from 1 of the overlap of the optimized PEPS with the initial state. Physically $\mathcal{I}(t)$ can be seen as a quantitative measure of the "distance" of the best PEPS from the exact time-evolved state. This quantity should be compared to a reference $\mathcal{I}_0(t)$ obtained by replacing $|\Psi(t)\rangle$ by the initial state i.e. $\mathcal{I}_0(t) = 1 - |\langle \exp(-iHt) \rangle_{\mathrm{RVB}}|$. A fit of the numerical data shows that $\mathcal{I}_0(t) \simeq \exp(-Ct^{-2})$, becoming exponentially small at small time. Nevertheless, the Cluster VO enables to gain several order of magnitude in $\mathcal{I}(t)$, e.g. for $t = 0.1$ we get $\mathcal{I}(t) \sim 10^{-6} \mathcal{I}_0(t)$. In fact we expect the optimization to become exact (on the cluster) in the $t \to 0$ limit, meaning $\mathcal{I}(t)/\mathcal{I}_0(t) \to 0$. At the same time finite size effects (FSE) should also disappear asymptotically since the Lieb-Robinson theorem [40] states that, after the quench, correlations propagate with a bounded velocity. So we believe the cluster method becomes conceptually exact in the small time limit. For increasing time, the optimization deteriorates a bit (as expected since the entanglement of the time-evolved state grows) but remains still quantitatively very good, e.g. we get $\mathcal{I}(t) \sim 10^{-3} \mathcal{I}_0(t)$ for $t = 0.5$. For comparison, we also show the infidelity for finite-size PEPS on the $4 \times 4$ torus using the site tensors obtained in the SU and EBVO methods (with the same virtual space). The infidelity of the SU PEPS is clearly a few order of magnitude higher that the one of the CVO. The EBVO is doing better than SU for $t > 0.15$ but still with an infidelity an order of magnitude (i.e. roughly $\times 10$) higher than the CVO for $t \sim 0.4 - 0.5$. However, this does not necessarily implies any hierarchy in the relevance of the various frameworks for the *infinite lattice* since, for this perspective, the CVO is subject to finite size effects.

Note that the CVO has been performed within two different PEPS manifolds using either U(1) or $\mathbb{Z}_2$ gauge-symmetric tensors. It has been argued (using TS decomposition) that, in the thermodynamic limit, the $U(1)$-gauge symmetry of the initial state is preserved during time evolution provided the quench Hamiltonian only acts on NN bonds. The $\mathbb{Z}_2$ gauge-symmetric PEPS are obtained by adding 3 extra tensors to the U(1) local tensor basis. Hence, the $\mathbb{Z}_2$ manifold is larger and includes the U(1) manifold, and we expect the optimization to give a better (smaller) overlap (infidelity). We remind however that, in the SU and EBVO frameworks, the weights of the 3 additional tensors were systematically found to vanish [27]. In contrast, Fig. 1 shows that, on a finite cluster, enlarging the PEPS manifold from U(1) to $\mathbb{Z}_2$ always lead, after optimization, to a larger overlap (or smaller infidelity) with the exact finite-size time-evolved state. Whether it is a finite size effect or the $\mathbb{Z}_2$ tensors are also relevant in the thermodynamic limit shall be discussed later on. The two procedures will be denoted as U(1)-CVO and $\mathbb{Z}_2$-CVO in the following, for convenience.

### 2.3.2 Physical observables

The above cluster optimization scheme provides a site tensor which can be used to construct a translationally invariant iPEPS. Using the CTMRG algorithm described above, this enables to compute various physical quantities on the infinite lattice (such as the energy density or the entanglement entropy) to be compared directly to the two other methods. As all methods, the CVO is fundamentally limited by the truncation of the PEPS ansatz to $D = 6$ whenever hitting the "entanglement barrier" after some typical time. Another source of error is the FSE which limits the accuracy of the optimization procedure. The computation of the observables however is not subject to further FSE since performed in the thermodynamic limit using the optimized U(1) or $\mathbb{Z}_2$ site tensors.

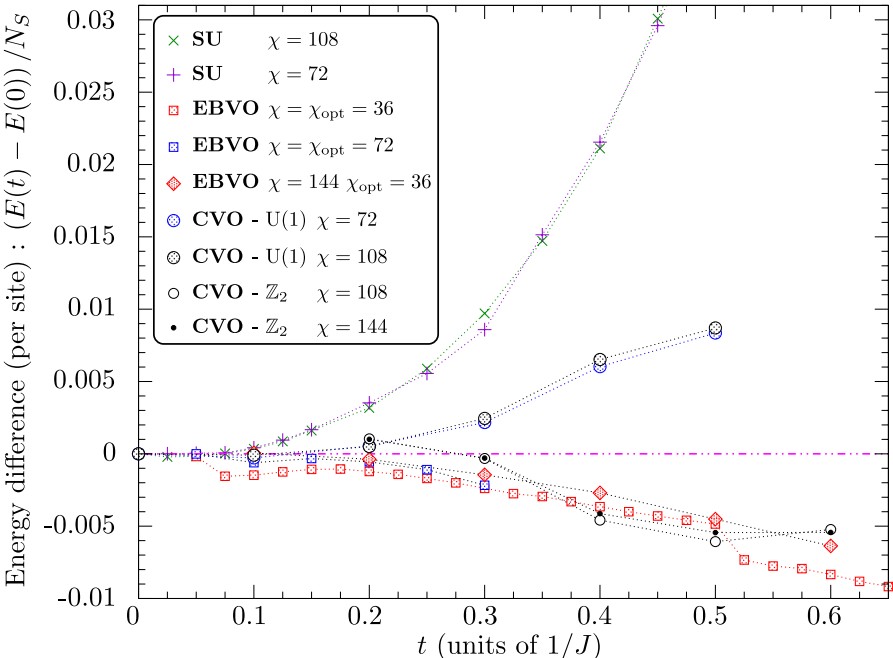

Figure 3: Energy difference (w.r.t. the $t = 0$ initial value) in the thermodynamic limit obtained with the embedded-bond and cluster VO methods and compared to the SU results. Difference iPEPS environment dimensions $\chi$ have been used as indicated in the legend. For the EBVO method, $\chi_{opt} = 36$ and $\chi_{opt} = 72$ have been used in the optimizations with a TS step $\tau = 0.025$ and $\tau = 0.05$, respectively (and the initial tensor is chosen as the SU tensor obtained at $t = \tau$). Computations have been done with $\chi = \chi_{opt}$ or $\chi = 144$.

# 3 Results

## 3.1 Energy density

We now turn to the results obtained by using the two optimization methods which we would like to compare to our previous SU results. The evolution of our closed system is unitary so that its energy should be a constant of motion, hence offering a simple test of the accuracy of our procedures. The energy deviation (per site) w.r.t. its initial $t = 0$ value plotted in Fig. 3 shows that the EBVO scheme provides a very significant improvement w.r.t the simple SU scheme. However such an improvement is obtained at the price of being more expensive in computer time. Also we observe that the symmetric CTMRG (used to compute the environment at each TS step) becomes unstable (or is subject to small spontaneous SU(2)-symmetry breaking) above some characteristic time, $t \gtrsim 0.55$ ($t \gtrsim 0.31$) for $\chi_{opt} = 36$ ($\chi_{opt} = 72$). Hereafter, EBVO computations will be done using $\tau = 0.025$, $\chi_{opt} = 36$ and $\chi = 144$, providing the best results.

Let us now move to the CVO results reported also in Fig. 3. Here we have performed optimizations within the $4 \times 4$ PEPS manifolds constructed from both U(1) and $\mathbb{Z}_2$ classes of site tensors. From the optimized site tensor one can then construct the iPEPS to compute the energy density in the thermodynamic limit. We find that the deviation of the energy density w.r.t. its initial value is remarkably small both in the U(1)-CVO and $\mathbb{Z}_2$-CVO procedures, but with a different sign. While the optimization itself is always well behaved, the symmetric CTMRG using the optimized site tensor to compute observables seems to become unstable,

as for the EBVO method, beyond some typical time of order $t \sim 0.55$. In any case, we find that all our variational methods show a much smaller energy deviation compared to the SU procedure, a first sign pointing for a better accuracy.

## 3.2 Transfer matrix spectrum and asymptotic properties

In this subsection we investigate the long-distance asymptotic limit. We argue that physical properties (like correlations) should not be modified in this limit, i.e. outside of some "light cone", during time evolution. We confront this statement to our calculations, using the TM as a tool to access the asymptotic limit. We show that the deviations from the expected time-invariance of the asymptotic correlations remain very small in the VO methods.

### 3.2.1 Lieb-Robinson bound

It is known from Lieb and Robinson's work [40] that, after the quench, the rate at which the information can propagate is bounded by the Lieb-Robinson (LR) velocity $v_{LR}$, providing the notion of a light-cone. As a consequence, there is a finite speed at which correlations and entanglement can be distributed [41–43]. These notions have been extended to the case where the initial state has power-law decaying correlations [44, 45]. The existence of a LR bound means that, after a finite time $t$, any correlation function $C(r, t)$ in the time-evolved state $|\Psi(t)\rangle$ can only be modified (in comparison to the $t = 0$ initial state) at finite distance $r \lesssim v_{LR}t$, apart from exponentially small tails. It means that the correlation $C(r, t)$ should retain its character, critical or short-range. In the second case, the finite correlation length characterizing the asymptotic behavior $r >> v_{LR}t$ should be time-independent.

### 3.2.2 Transfer matrix spectrum

The double layer $\langle \Psi(t)|\Psi(t)\rangle$ TN on the (let's say) upper infinite half-plane leads to a one-dimensional (1d) boundary which can be approximated by a Matrix Product States (MPS) defined by a single site tensor $T$ of finite bond dimension $\chi$ (same as the environment tensor obtained by CTMRG). Useful information – like any correlation function at all distances – is encoded in the $\chi^2 \times \chi^2$ transfer matrix $\mathcal{T} = T^{\otimes 2}$ obtained by contracting over the $D^2$ virtual indices (up to finite-$\chi$ errors). However, the *spectrum* of $\mathcal{T}$ only provides information on the correlations in the asymptotic $r \to \infty$ limit which should not be affected by finite-time propagation, from the existence of a bound in the velocity of the information spreading. Therefore, the invariance of the spectrum with time provides an additional precise test of the different methods. In the case of SU(2) symmetry, the TM eigenvalues can be labeled by their degeneracy $g = 2S + 1$, defining spin sectors. Investigating the deviations of the leading eigenvalues of the various SU(2) sectors, compared to their initial values, offers a stringent test.

### 3.2.3 Criticality

From the existence of the LR bound, we expect the critical (dimer) correlations in the asymptotic limit ($r \to \infty$) to be robust during the time evolution. This can be tested by examining the gap between the leading and subleading eigenvalues of the TM. The TM spectra obtained in the initial NN RVB and, for $t = 0.5$, in the SU and EBVO methods are shown in Fig. 4 for increasing environment dimension $\chi$. Additional data for spectra obtained using CVO are shown in Appendix A. All results look very much alike and are consistent with a vanishing singlet gap (defined by the difference between the leading and subleading $g = 1$ eigenvalues) in the limit $\chi \to \infty$ characteristic of a critical phase.

For a more quantitative analysis we have computed the maximum correlation length from the ratio of the first two leading eigenvalues $\lambda_0 = 1$ and $\lambda_1 < 1$, $\xi_{max} = -1/\ln(\lambda_1/\lambda_0)$.

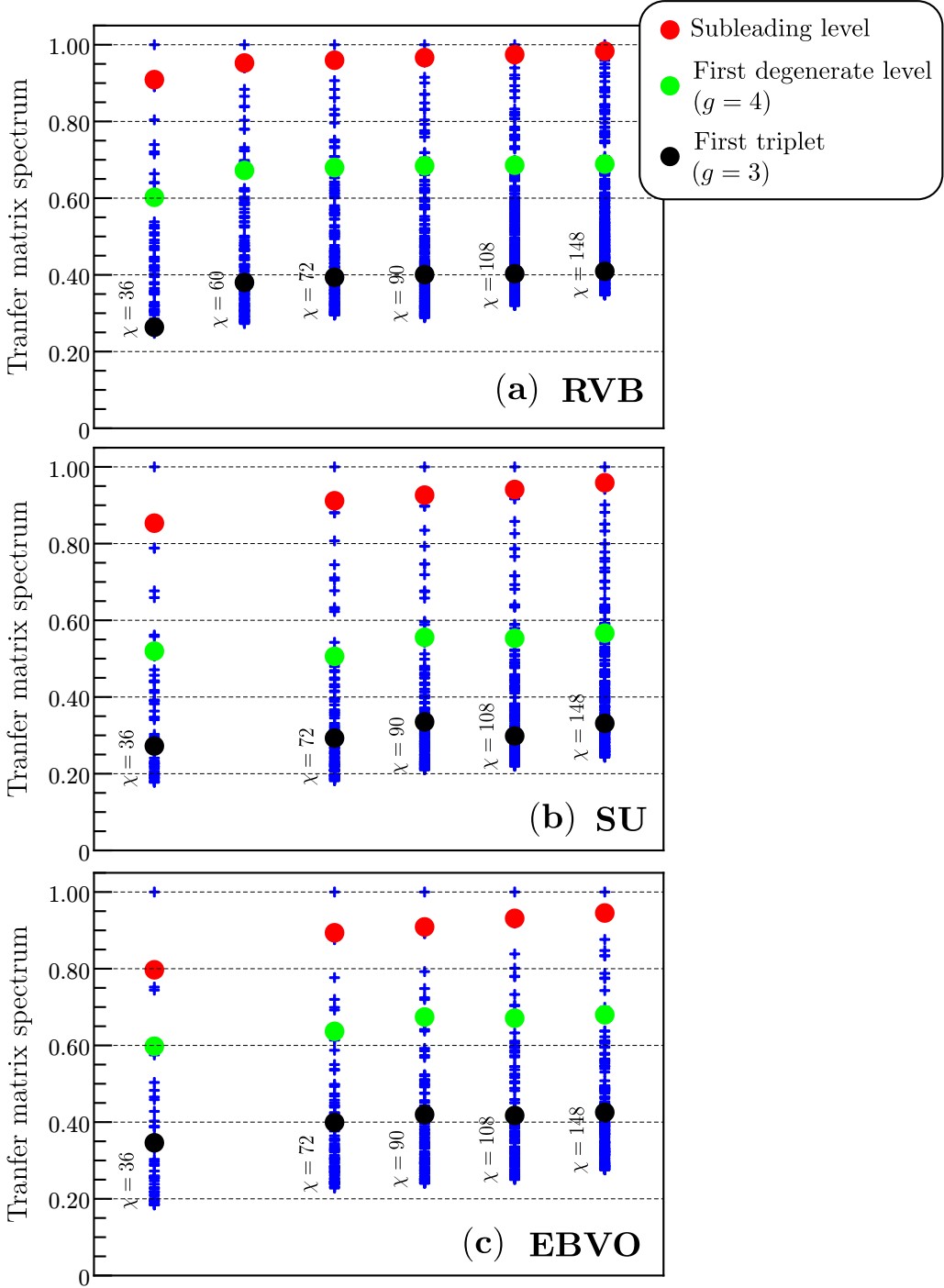

Figure 4: TM spectra in the initial RVB state **(a)** and at time $t = 0.5$ **(b-c)** for different values of $\chi$. The finite-time spectrum has been computed using the SU and EBVO methods. Only the largest 40 eigenvalues are shown, normalized such that the leading one is set to 1. Different symbols are used to highlight particular levels, the subleading eigenvalue ($g = 1$), and the largest eigenvalues with degeneracy $g = 4$ and $g = 3$. Note that the $g = 3$ multiplet of the NN RVB state is exactly degenerate with $g = 2, 4$ and 6 multiplets. Note also that some SU data have already been reported in [27].

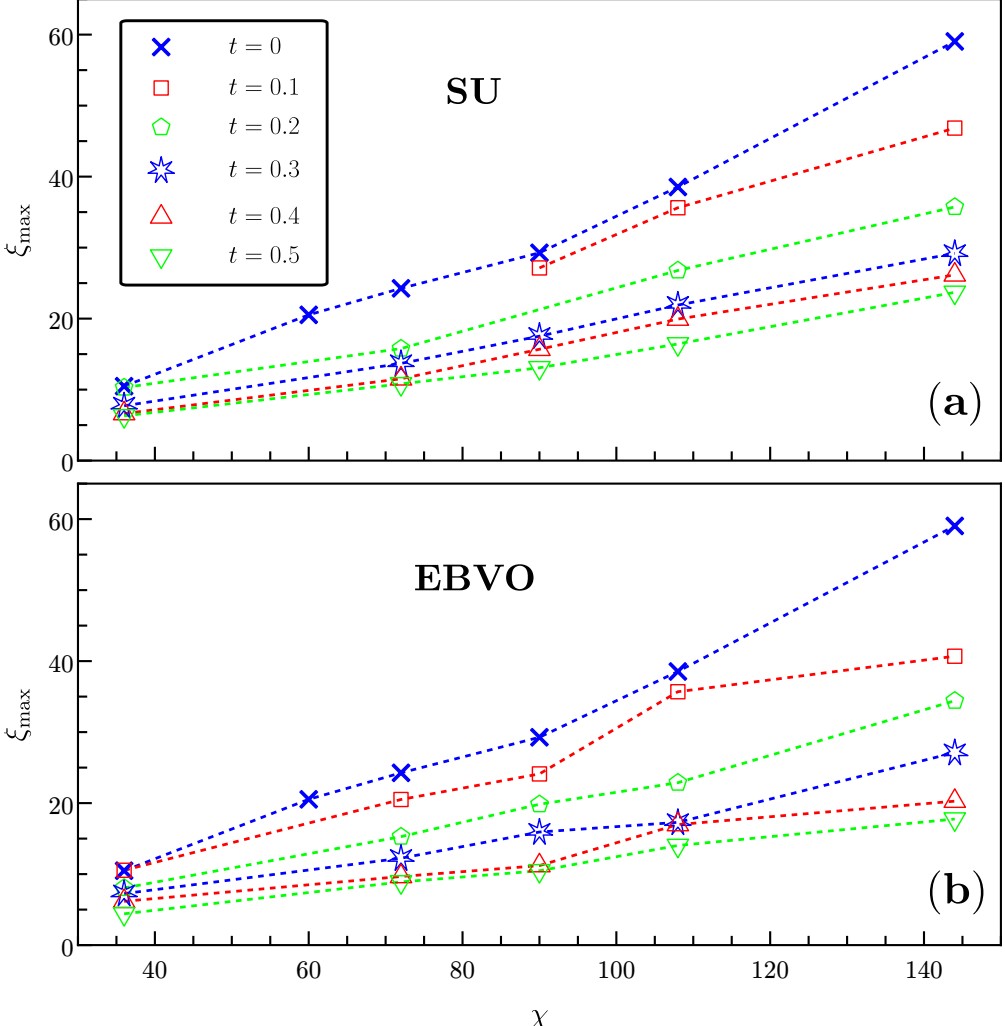

Figure 5: . Maximum correlation length plotted as a function of the environment dimension $\chi$, obtained in the SU method (panel **(a)** - data from Ref. [27]) and in the EBVO method (panel **(b)**) for different time $t$ (see legend). A steady increase of $\xi_{\max}$ with $\chi$ is observed with no sign of saturation.

The results are shown in Fig. 5 and in Appendix A, revealing the absence of saturation with $\chi$ for all methods. A diverging (maximum) correlation length is indeed consistent with the preservation of the asymptotic critical nature of the (dimer) correlations under time evolution. We observe that, quite generically, $\xi_{\max}(t, \chi) \propto \chi$. However we note that the slope $s_{\xi}(t) = \partial \xi_{\max}(t, \chi)/\partial \chi$ is significantly reduced for increasing time. E.g., at $\chi = 144$, maximum correlation lengths $\sim 20$ are obtained for $t = 0.5$, compared to $\sim 60$ at $t = 0$. We believe the change with time of the finite-$\chi$ corrections is not inconsistent with the LR bound argument.

In addition to the critical nature of the bulk, we expect the boundary MPS to be described, for any finite time, by the same Conformal Field Theory (CFT) of central charge $c = 1$ as for the initial NN RVB state. To verify this important feature we have computed the MPS Von Neumann entanglement entropy (partitioning the infinite chain into two halves). A selection of the data is plotted in Fig. 6 as a function of the logarithm of $\xi_{\max}$. The data are found to be consistent with a linear scaling $S_{\text{vN}} = \frac{c}{6} \ln \xi_{\max}$ [46, 47], even though the range of variation of $\ln \xi_{\max}$ shrinks significantly for increasing time making the comparison to the CFT prediction less precise.

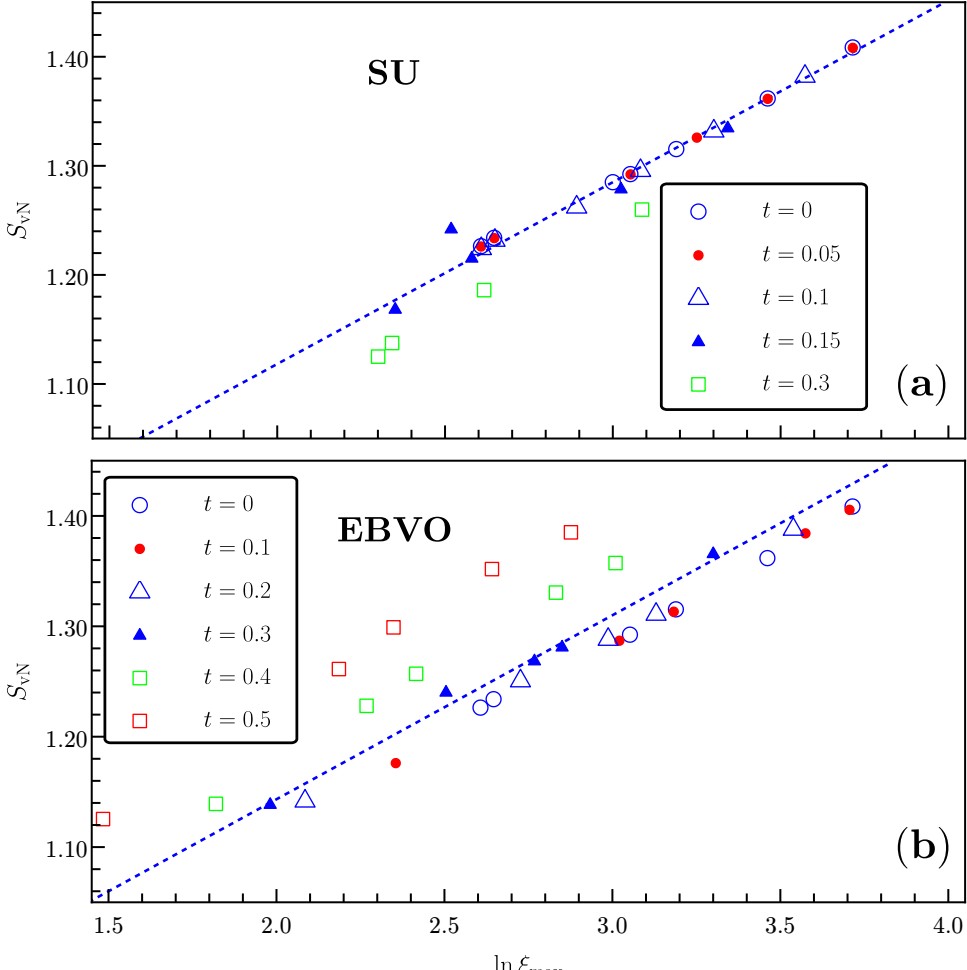

Figure 6: Von-Neumann entanglement entropy of the 1d boundary theory as a function of $\ln \xi_{\mathrm{max}}$ for various values of the time $t$ (see legend). The dashed line is a guide to the eye showing the expected CFT scaling with central charge $c = 1$. The SU (**a**) and the EBVO (**b**) schemes have been used. Both sets of data are compatible with $c = 1$.

### 3.2.4 Finite correlation lengths

Despite the existence of critical dimer correlations, many correlations remain short-range like the spin-spin correlation. The LR velocity bound implies that such correlations should not depend on time asymptotically at long distance $r > v_{\mathrm{LR}}t$. In other words, the maximum correlation length associated to a given observable should be time invariant. Interestingly, the TM provides information about these asymptotic (finite) correlation lengths which can be distinguished by the degeneracy $g$ of the corresponding eigenvalues. As shown in Appendix A, the spectra of all $g > 1$ eigenvalues is gapped in the $\chi \to \infty$ limit, in contrast to the gapless $g = 1$ (singlet) spectrum (associated to the dimer critical correlations). Since in that limit all spectra become dense, one way to compare TM spectra at different times is to compare their gaps $1 - \lambda^{(g)}(t)$ or, equivalently, their associated leading correlation lengths $\xi^{(g)}(t) = -1/\ln(\lambda^{(g)}(t))$, where $\lambda^{(g)}$ is the leading eigenvalue of the $g$-degenerate eigenvalue spectrum.

Let us first start with the reference NN RVB state (at $t = 0$). In the top panel of Fig. 4, we have identified in the TM spectrum (apart from the gapless singlet spectrum discussed above) the smallest gaps of the $g = 4$ and $g = 15$ eigenvalues, connected to the largest

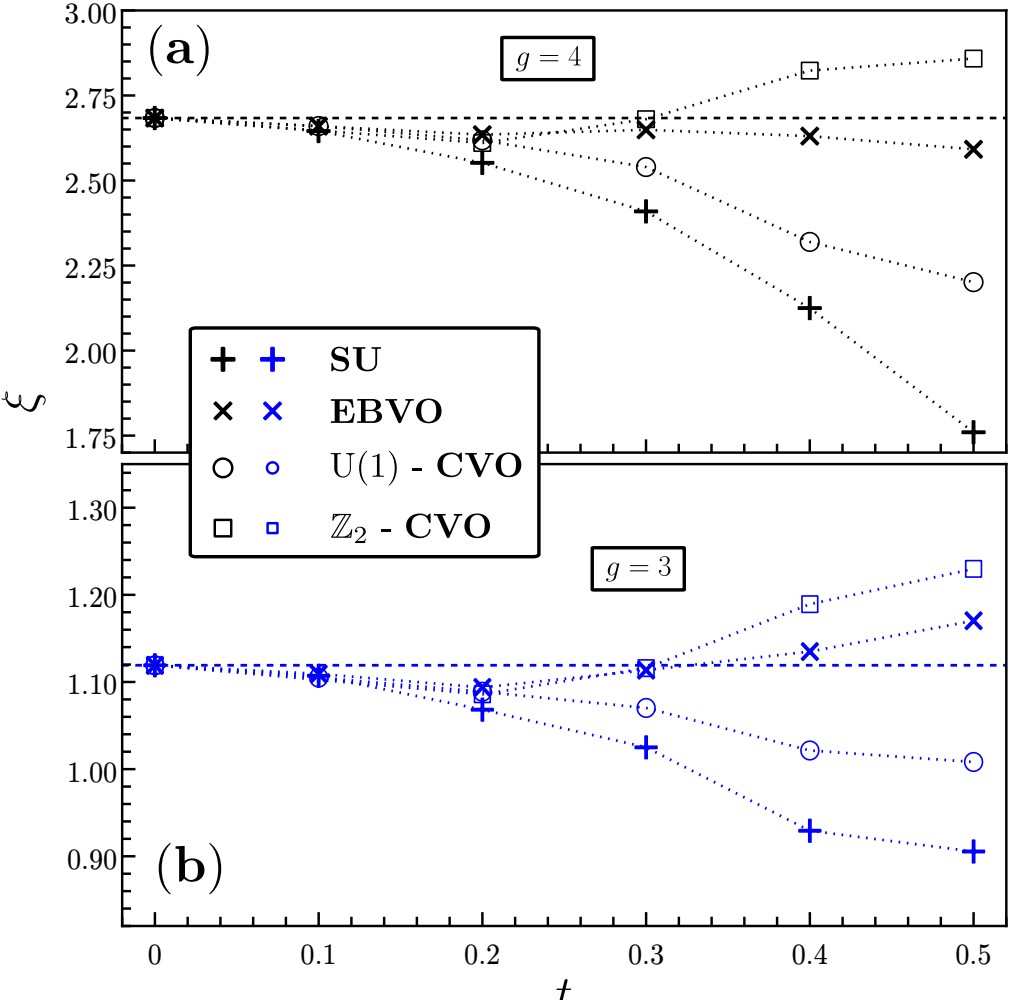

Figure 7: Asymptotic correlation lengths computed at $\chi = 144$ are plotted versus time. All correlation lengths are labeled by the degeneracy of the corresponding TM eigenvalues and are associated to different operators. SU (crosses), EBVO (diagonal crosses), U(1)-CVO (circle) and $\mathbb{Z}_2$-CVO (squares) results are shown. Panel **(a)**: leading correlation length $\xi^{(4)}$ associated to the gap of the $g = 4$ TM spectrum and corresponding to the spinon correlations [27]. Panel **(b)**: leading correlation lengths $\xi^{(3)}$ associated to the gaps of the $g = 3$ TM spectra (originating from the $g = 15$ largest eigenvalue of the $t = 0$ TM spectrum). Note that $\xi^{(3)}$ corresponds to the spin-spin correlation length [27].

range correlations in the system (apart from the critical dimer correlations). The large $g = 15$ degeneracy reflects the very special fine-tuned nature of the NN RVB with different operators exhibiting identical (asymptotic) correlations. In fact, at small time $t$, the $g = 15$ levels are split into four almost degenerate levels with $g = 2, 3, 4, 6$, as a result of the approximate nature of the time-evolved state. Note that, from previous work [27], we know that the $g = 3$ spectrum corresponds to the spin-spin correlation function. From the bottom panels of Fig. 4 we see that the TM spectra, in particular the $g = 4$ and $g = 3$ gaps, do not change very much at finite time, $t = 0.5$.

The deviations of the corresponding leading correlation lengths $\xi^{(g)}(t)$, $g = 4, 3-$ shown in Fig. 7 using $\chi = 144$ – w.r.t. the ones of the NN RVB $\xi^{(g)}(0)$ give another quantitative measure of the accuracy of our methods, in addition to the energy conservation check. Again,

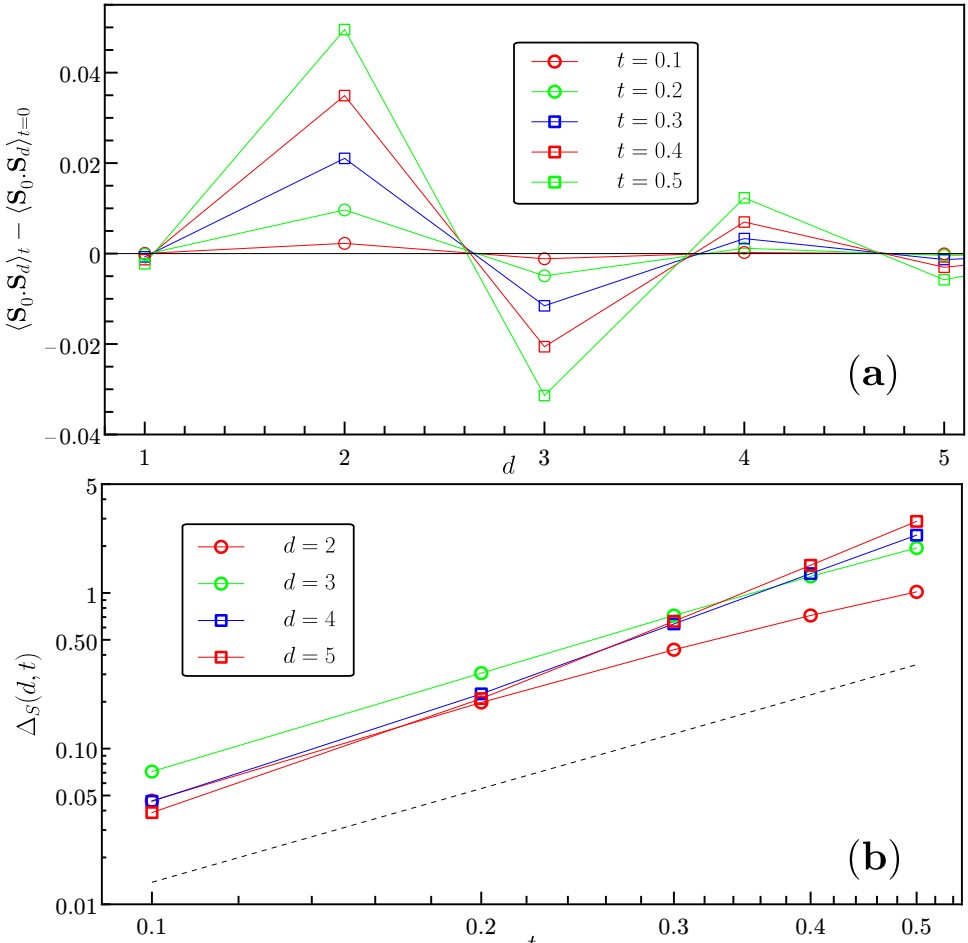

Figure 8: (a) Spin-spin correlations $C_S(d,t) = \langle \mathbf{S}_0 . \mathbf{S}_d \rangle_t$ (w.r.t. the $t = 0$ values) along the horizontal (or vertical) axis up to distance $d = 5$, for $t = 0.1, 0.2, 0.3, 0.4, 0.5$. Tensor optimizations have been done in the EBVO scheme ($\tau = 0.025$, $\chi_{\text{opt}} = 36$) and the correlations have been computed in the thermodynamic limit with $\chi = 144$. (b) Relative change of the *staggered* spin-spin correlation $\Delta_S(d,t)$ (see text for exact definition) versus time (log-log plot) for $d = 2, 3, 4, 5$. The dashed line corresponds to the $t^2$ behavior obtained for $d = 2$ on the $4 \times 4$ cluster: $\Delta_S(d = 2, t) \sim 1.385\, t^2$.

we observe that the VO methods give rise to smaller deviations than the SU method, consistently with the analysis of the energy density. Interestingly, we note that the sign of the (small) correlation length deviations, an artifact of our approximate treatments, depends on the method.

## 3.3  Finite distance correlations

Although the asymptotic behavior should not be affected for increasing time $t$, correlations at finite distances should get stronger. As an example we investigate the spin-spin correlations $C_S(d,t) = \langle \Psi(t) | \mathbf{S}_i \cdot \mathbf{S}_{i+d} | \Psi(t) \rangle$ between two sites along one of the crystal axis of the square lattice, separated by a short distance $d$. Figure 8(a) shows the variations of the correlations $C_S(d,t) - C_S(d,0)$ for increasing time up to $t = 0.5$. Here a EBVO optimization procedure followed by an iPEPS/CTMRG computation using $\chi = 144$ was used. We observe a rapid increase of the antiferromagnetic short-distance correlations, reaching at $t = 0.5$ and distances

$d \sim 3-5$ of the order of 150 to 300 % of their initial values. Similar numerical studies have also been performed in the context of experiments on 2D arrays of Rydberg's atoms [48]. We found that correlations increase at short-time as $t^2$ for all distance $d > 1$, in contrast to the $t^{2+4d}$ behavior found in Ref. 48. We argue in Appendix B that this qualitative difference is due to the existence of finite correlations in the initial state while a product state was considered instead in Ref. 48.

To be more quantitative we have defined the relative increase as $\Delta_S(d,t) = (\tilde{C}_S(d,t) - \tilde{C}_S(d,0))/\tilde{C}_S(d,0)$, where the antiferromagnetic oscillations of $C_S(d,t)$ are absorbed by defining $\tilde{C}_S(d,t) = (-1)^d C_S(d,t) > 0$. $\Delta_S$ is plotted in Fig. 8(b) as a function of time, using logarithmic scales on both axis. Note that we restrict here to short distances, typically below $d \sim 5$ since, beyond that, we enter the asymptotic regime $d >> 1$ which is governed primarily by the TM spin-spin correlation length, $C_S(d,t) \propto \exp(-d/\xi^{(3)}(t))$. Since (i) due to our approximate scheme, $\xi^{(3)}(t)$ deviates slightly from the initial value $\xi^{(3)}(0)$ of the NN RVB state and (ii) it concerns a regime of very small magnitudes of correlations $\tilde{C}_S(d,t) < 10^{-5}$, no qualitative analysis can be done for $d \geq 5$.

The $t^2$ behavior can be understood from a simple (crude) argument based on the LR bound, from which we can also estimate some LR velocity. For a given distance $d > \xi^{(3)}$, at (sufficiently) short-time one can assume to be outside of the LR light-cone where correlations are non-zero from the very beginning of the time evolution. In that region, the correlation function in fact exhibits "leaks" with spatio-temporal exponential decay [45],

$$C_S(d,t) \simeq K \left( \exp[-(d - v_{LR}t)/\xi^{(3)}] + \exp[-(d + v_{LR}t)/\xi^{(3)}] \right).$$

Hence, in that limit, the relative increase of the correlations becomes

$$\Delta_S(d,t) \simeq \cosh(v_{LR}t/\xi^{(3)}) - 1 \sim \frac{1}{2}(v_{LR}t/\xi^{(3)})^2.$$

From the numerical estimation of the $t^2$ coefficient for $d = 3$ we estimate $v_{LR} \simeq 4.1$ in unit of $J$.

## 4 Conclusions

In summary, we have applied tensor network techniques to investigate the time evolution of a 2D (critical) spin liquid on the square lattice after a sudden quench. The quench Hamiltonian is the simple NN Heisenberg model so that all symmetries of the initial state (invariance under space group and spin-rotation symmetries) are preserved during the time evolution. Practically, this allows to represent the time-evolving state by a translationnally invariant singlet PEPS defined by a single symmetric site tensor. The time-evolution of the state is therefore simply encoded in the time-evolution of the site tensor. Using a basis of the local site tensors (of a given virtual bond dimension $D = 6$), the (highly non-linear) optimization problem hence translates in finding the best linear combination of the basis tensors. The U(1) gauge symmetry of the initial state plays also a special role conferring automatically critical dimer correlations to the time-evolving state. The U(1) gauge symmetry is enforced by construction in the U(1)-CVO method and preserved by the application of the two-site gate in the SU and EBVO frameworks. Nevertheless, it is no longer explicit at the level of the on-site tensor in the $\mathbb{Z}_2$-CVO method. In that case, since the PEPS ansatz seems to also remain critical under time evolution, one possibility is that the U(1) gauge symmetry is somehow "hidden". Another possibility would be that the breaking of the gauge symmetry from U(1) to $\mathbb{Z}_2$ would be a finite size effect of the optimization on a finite cluster.

We have tested the respective accuracy of our methods by different means. First, as expected in the case of a unitary evolution, the energy (i.e. the expectation value of the quench Hamiltonian in the time-evolving state) should be conserved. The observed deviation of the energy (per site) remains quite small in the VO methods – typically smaller than 1.5% for $t \leq 0.5$ – while it rises to around 8% in the SU method at $t = 0.5$.

We have also considered the consequences of the Lieb-Robinson theorem stating an upper-bound of the velocity at which information propagates. From this theorem, there exists a "light-cone" beyond which correlations should remain unchanged (apart from exponentially small tails), imposing some constraints on the properties of the time-dependent TM which governs all asymptotic properties. In particular, we have investigated the criticality of the system, finding it is preserved under time-evolution in all methods. In particular, the system boundary is found to be described by the same Conformal Field Theory of central charge $c = 1$ during time evolution. To be more quantitative, we have also studied the finite correlation lengths associated to spinon and spin correlations. In our approximate schemes small deviations are of course expected but we found the latter remain quite small, especially in the VO methods, up to accessible times of the order of $0.5 - 0.6$.

To investigate how correlations develop at finite distances we have considered spin-spin correlations which are short-ranged in the initial NN RVB state and, therefore, are simpler to analyse. Considering spacio-temporel parameters outside of the (potential) light-cone the rate of spreading of correlations is estimated.

Finally, let us describe the pros and cons of the various methods we have used. For all of them the main limitation is of course the finite virtual space dimension $D = 6$ that limits the bond entanglement to $\ln 6$ while the latter is known to increase indefinitely with time. For longer times than studied here, it would be therefore necessary to increase the bond dimension, adding some extra SU(2) multiplet(s) to the virtual space. Even though the "entanglement wall" limits all methods, it is nevertheless meaningful to compare the methods at small times. We have provided arguments that the VO methods, although much more costly in CPU-time, provide a higher accuracy, according to the criteria mentioned above. Note that the source of errors of the different VO methods are clearly different: while the optimization suffers from finite size effects in the CVO, the tensor update remains essentially local in the EBVO (although taking into account the environment). Let us also mention that, for all methods, the computation of observables on the infinite 2D lattice relies on a (symmetric) CTMRG procedure. Generically we experience some instability issues for $t > 0.5$ (typically spin-rotation symmetry breaks down spontaneously). It may signal the fact that the iPEPS deviates too much from the true time-evolved state and is no longer physical.

Lastly, we would like to point out that our methods are quite versatile and can be applied to other lattices/type of spin liquids/quench Hamiltonians. In particular, topological short-range NN RVB on non-bipartite lattices could be studied with the same methods. Note that adding longer range interactions (like a next-NN frustrating Heisenberg coupling) to the quench Hamiltonian is an easy task in the CVO framework. Interestingly, such variational methods could be well suited to investigate Floquet dynamics.

# A  Transfer matrix spectra in the CVO methods

For completeness we provide in this Appendix results on the TM spectra obtained using the CVO method – to get the PEPS site tensor – followed by a CTMRG procedure to obtain the fixed-point MPS boundary. We focus here on the finite-$\chi$ effects.

A comparison of the $t = 0.5$ spectra using Cluster Variational Optimization within the restricted U(1) **(a)** of full $\mathbb{Z}_2$ tensor basis **(b)** is shown in Fig. 9, for increasing $\chi$ values. These

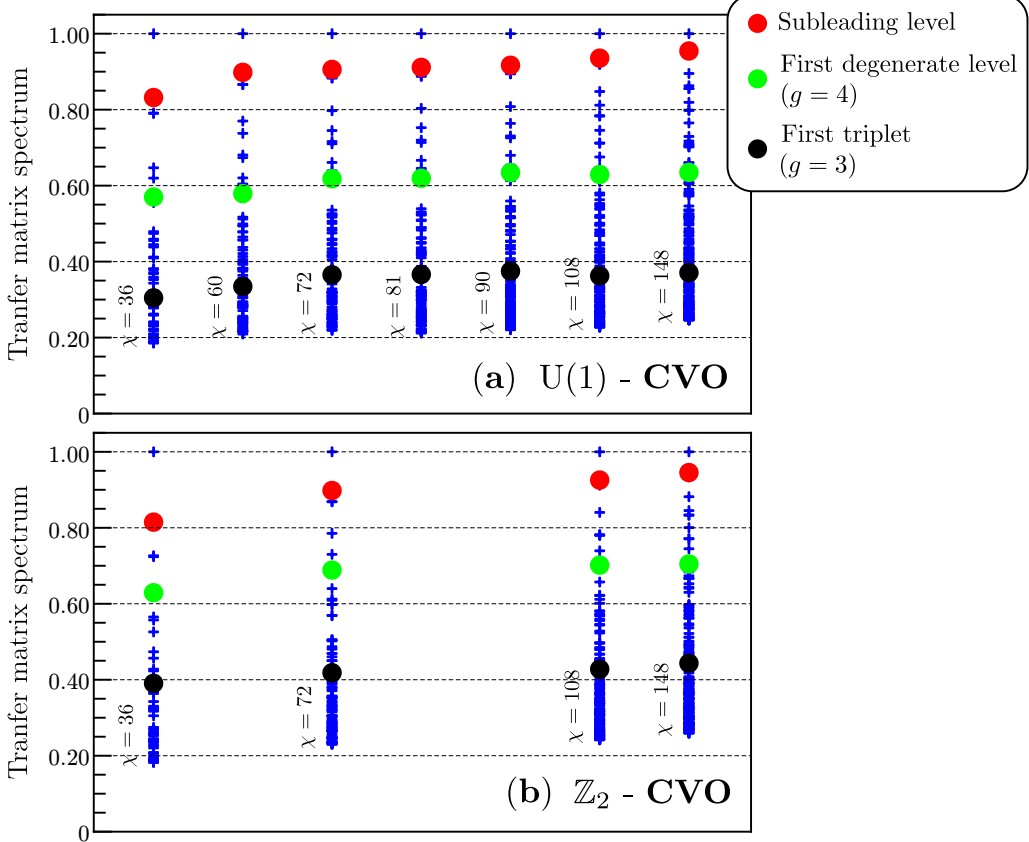

Figure 9: TM spectra at time $t = 0.5$ for different values of $\chi$ computed using Cluster Variational Optimization within the restricted U(1) **(a)** or full $\mathbb{Z}_2$ **(b)** tensor basis. Only the largest 40 eigenvalues are shown, normalized such that the leading one is set to 1. Different symbols are used to highlight particular levels, the subleading eigenvalue ($g = 1$), and the largest eigenvalues with degeneracy $g = 4$ and $g = 3$.

spectra resemble very much those obtained in the SU and in the EBVO methods shown in the main text. In particular, the data are consistent with a vanishing gap in the $\chi \to \infty$ limit although with finite gaps associated to spinon (leading $g = 4$ multiplet) and spin-spin (leading $g = 3$ multiplet) correlations.

The maximum correlation lengths extracted from the TM gaps plotted in Fig. 10 as a function of the environment dimension $\chi$ show the same linear behaviors as those obtained within the SU or EBVO methods, suggesting again a critical state at all times. It is natural to expect such a critical behavior in the U(1)-CVO method since the ansatz bears the same U(1) gauge symmetry as the initial NN RVB state. However, interestingly enough, this gauge symmetry is absent at the level of the $\mathbb{Z}_2$ site basis tensors and, therefore, may be hidden in the ansatz obtained by the $\mathbb{Z}_2$-CVO method.

For completeness we also show in Fig. 11 the behavior of the boundary MPS entanglement entropy versus the logarithm of the maximum correlation length. The results are again very similar to the ones of the SU and EBVO methods, in agreement with the linear scaling expected in a $c = 1$ central charge CFT.

Lastly, we show in Fig. 12 the leading correlation lengths of the NN RVB state and of the time-evolved state at $t = 0.5$ plotted as a function of $1/\chi$. The time-evolved state is computed here using the U(1)-CVO scheme. The data reveal moderate finite-$\chi$ effects with small linear $1/\chi$ corrections in the asymptotic $\chi \to \infty$ limit corresponding to the exact contraction of the

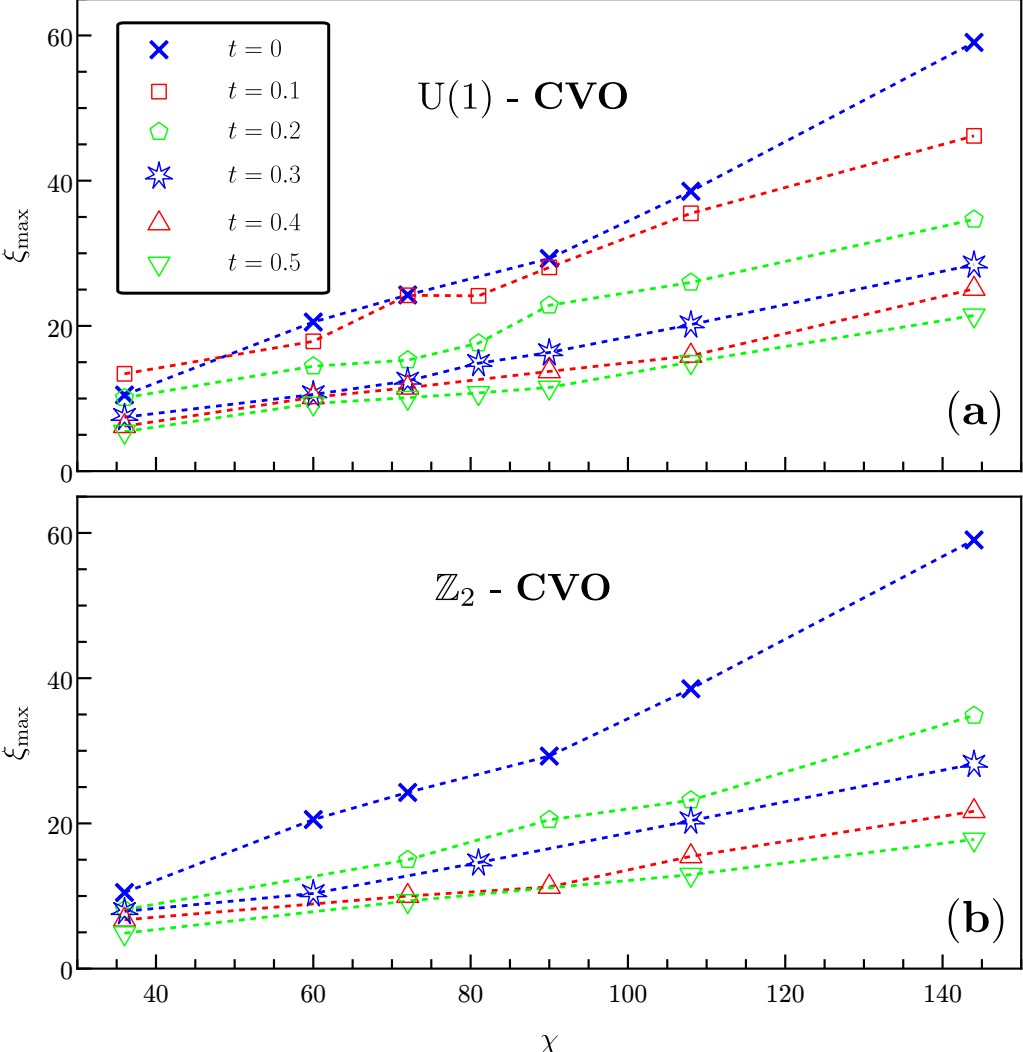

Figure 10: Maximum correlation length plotted as a function of the environment dimension $\chi$, obtained in the U(1) (panel **(a)**) and $\mathbb{Z}_2$ (panel **(b)** CVO methods. The different symbols correspond to different values of the time $t$ (see legend).

iPEPS tensor network. We see that the data obtained with the largest available $\chi = 144$, as mostly reported in the text, are already accurate. The same conclusion applies to the data obtained by all other methods.

## B  Short-time expansion of the spin-spin correlations

Here we estimate the short-time bevavior of the spin-spin correlation $C_S(d,t)$ where $d$ is the linear (or Manhattan) distance between two sites $i$ and $j$, $d > 1$. Expanding the correlation function $C_S(d,t) = \left\langle \exp(iHt)\mathbf{S}_i \cdot \mathbf{S}_j \exp(-iHt)\right\rangle_0$ to second order in $t$, $C_S(d,t) \sim C_S(d,0) + C_S^{(1)}(d,t) + C_S^{(2)}(d,t)$, one gets

$$C_S^{(1)}(d,t) = -i\,t \left\langle [\mathbf{S}_i \cdot \mathbf{S}_j, H] \right\rangle_0, \tag{B.1}$$

$$C_S^{(2)}(d,t) = -\frac{t^2}{2} \left\langle [[\mathbf{S}_i \cdot \mathbf{S}_j, H], H] \right\rangle_0, \tag{B.2}$$

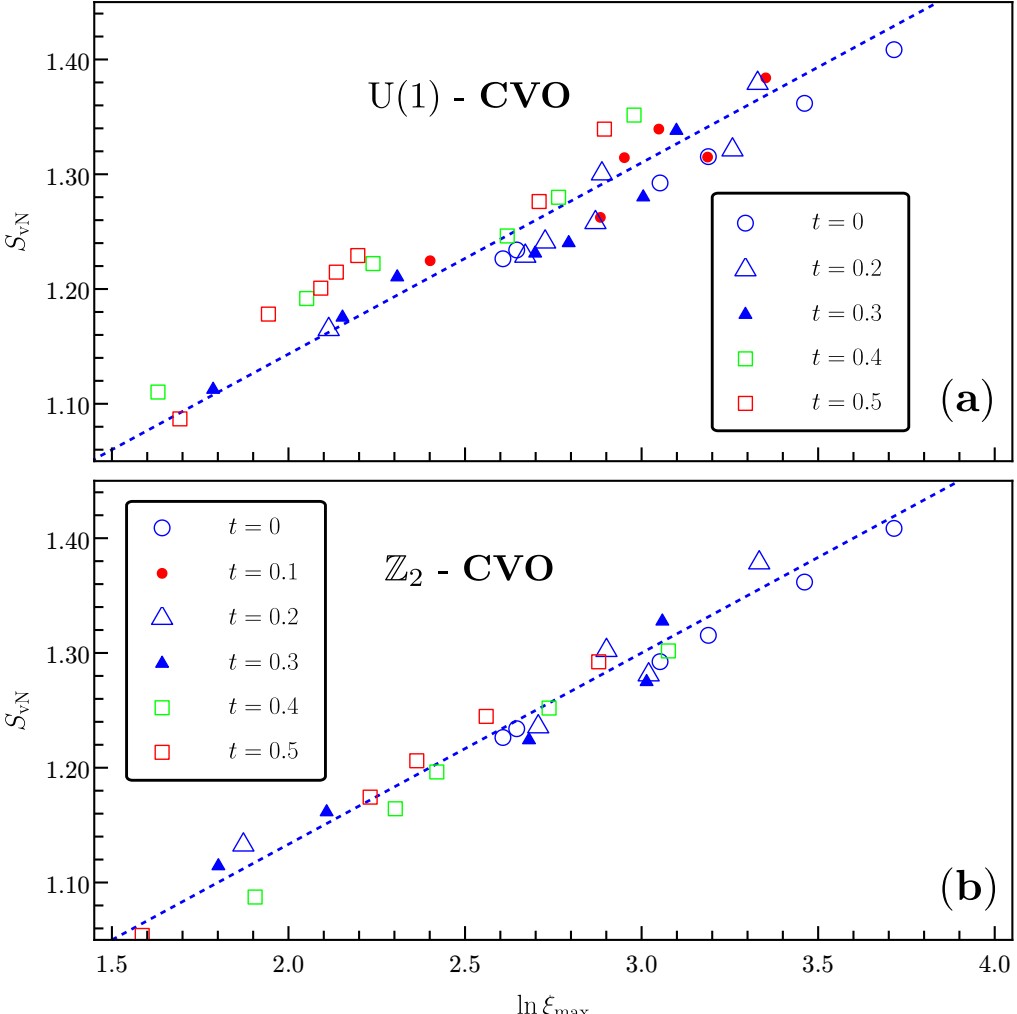

Figure 11: Von-Neumann entanglement entropy of the 1d boundary theory as a function of $\ln \xi_{\max}$ for various values of the time $t$ as shown in the legend. The dashed line is a guide to the eye showing the expected CFT scaling with central charge $c = 1$. The U(1) (**a**) and the $\mathbb{Z}_2$ (**b**) CVO schemes have been used.

where $\langle \dots \rangle_0$ is the expectation value taken in the initial state $|\Psi_0\rangle$. Using the expression of the Heisenberg hamiltonian $H$, the commutator $O_{ij}^{(1)} = [\mathbf{S}_i \cdot \mathbf{S}_j, H]$ can be written as,

$$O_{ij}^{(1)} = \sum_{k(i)} \mathbf{S}_j \cdot (\mathbf{S}_i \times \mathbf{S}_k) + \sum_{p(j)} \mathbf{S}_i \cdot (\mathbf{S}_j \times \mathbf{S}_p),$$

where $k(i)$ and $p(j)$ are NN sites of $i$ and $j$, respectively. Note that this operator has a zero expectation value in the RVB state (which is invariant under time-reversal) so that the $t$-linear term vanishes in the short-time expansion. Although the exact analytic expression becomes complicated, the double-commutator in $C_S^{(2)}(d, t)$ has the following structure,

$$O_{ij}^{(2)} = G_i^{(3)} G_j^{(1)} + G_i^{(1)} G_j^{(3)} + G_i^{(2)} G_j^{(2)},$$

where $G_k^{(n)}$ is a $n$-spin operator defined on a finite support of size 2 including site $k$. Since all terms are invariant under time-reversal, $O_{ij}^{(2)}$ has generically a finite expectation value. This property has been checked numerically on the $4 \times 4$ torus for all available distances between

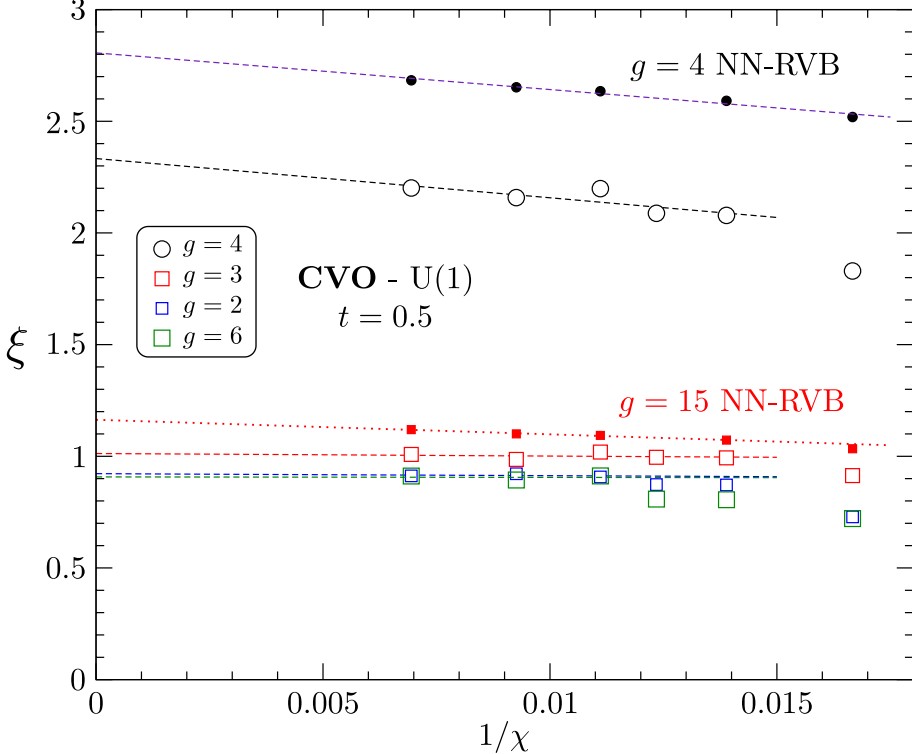

Figure 12: Scaling of various (finite) correlation lengths versus $1/\chi$. Data are obtained at $t = 0.5$ with the Cluster VO-U(1) method. The data obtained in the reference NN RVB state are shown as filled symbols. All correlation lengths $\xi^{(g)}$ are labeled by the degeneracy $g$ of the corresponding TM eigenvalues and, hence, correspond to different type of correlation functions (e.g. $g = 3$ corresponds to spin-spin correlations).

$i$ and $j$. Note also that this property will cease to be true if the expectation value is taken in a product state (like the classical Néel state) and the supports of the operator do not overlap, i.e. for $d > 4$.

## Acknowledgments

We acknowledge inspiring discussions with Mari-Carmen Banyuls, Andreas Läuchli, Norbert Schuch, Luca Tagliacozzo, and Frank Verstraete.

**Funding information** We acknowledge support from the TNTOP ANR-18-CE30-0026-01 grant awarded by the French Research Council. This work was granted access to the HPC resources of CALMIP center under the allocation 2022-P1231.

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
