# Peer review of "Tensor network variational optimizations for real-time dynamics: application to the time-evolution of spin liquids"

_SciPost Physics, doi:SciPost Phys. 15, 158 (2023)_

## Round 2 · Referee Report · Anonymous (Referee 2) · 2023-7-3

Strengths

1) contains a systematic analysis of an attempt to solve a very challenging problem obtaining a few interesting results in the process.

Weaknesses

1) results are limited to iPEPS bond dimension 6. It is not obvious how hard it would be to increase it within the employed machinery, and how would it affect the results.

Report

The paper considers the problem of numerical simulation of non-equilibrium quantum dynamics resulting from a sudden quench in a two-dimensional square lattice. The system starts in a specific Resonating Valence Bond state that gets evolved by the Heisenberg Hamiltonian. The state of the system is approximated using iPEPS ansatz on an infinite square lattice, where the authors employ SU(2)-symmetric, translationally, and rotationally invariant iPEPS representation with one-spin unit cell. The initial state has a compact representation of that form, and with the Hamiltonian conserving the symmetries, the main goal of the paper is to develop and test two procedures to approximately describe the time dynamics, preserving explicit iPEPS symmetries. One method is a variant of full-update scheme, and another follows cluster update, finding finite PEPS approximating exact evolution on a small cluster and embedding it on an infinite lattice. The paper is a direct extension of the previous work by the authors, where a simple-update algorithm was employed for the same problem.

The paper mostly focuses on the accuracy of the integration schemes, where the methods described here seem to greatly outperform the simple-update scheme. Two measures used for testing are conservation of energy and asymptotic correlation lengths (which should be time-independent due to the Lieb-Robinson bound). I find the discussion sufficiently self-standing, providing enough details to get an intuition about the limitations and perspectives of the method. An additional comparison with a method that does not explicitly enforces all the iPEPS symmetries would be quite educational to me, however, this seems out of the scope of the current material.

The article takes on an open problem of real-time evolution in a two-dimensional quantum system and, in my opinion, contains sufficient material to warrant publication. It provides a careful analysis of the numerical convergence of a very challenging problem, but also sneaks in a few physical results in the setup.

Requested changes

1) It is hard to qualitatively judge the data in Figs 4. and 9. I believe the presentation could be better here. Adding grid lines or some guide-for-an-eye lines might be helpful. Also putting all panels in a single row might aid in comparison - I don't think a linear x-scale is particularly meaningful here, so using values of \chi as labels might help to fit in all the data. Perhaps data in Figs 4. and 9 can be even combined into one figure if everything fits.

2) Results on the central charge in Fig. 6 seem to be lost in the abstract, introduction, or conclusion. It might be worth mentioning it somewhere beyond the main text.

3) Typos: -Last sentence of 2.2.2; I don't see M defined. -abbreviation FSE appears in 3rd paragraph of 2.3.1, but I see it defined only in 2.3.2 - "quantities quantities" in 2.3.2

  • validity: high
  • significance: good
  • originality: low
  • clarity: high
  • formatting: good
  • grammar: good

Author:  Ravi Teja Ponnaganti  on 2023-07-25  [id 3834]

(in reply to Report 2 on 2023-07-03)

We thank the referee for pointing out the strength of the work.

However the referee remarks that "results are limited to iPEPS bond dimension 6. It is not obvious how hard it would be to increase it within the employed machinery, and how would it affect the results."

Indeed, this is a legitimate concern. We would like to first mention that the bond dimension cannot be increased continuously but only by steps to comply with the singlet (SU(2)-symmetric) character of the wave function. Furthermore, looking back at Fig. 7(a) of the preceding paper PRB 106, 195132 (2022) suggests that D=8 (obtained by simply adding a single spin-1/2 virtual irrep is not really justified, a spin-3/2 irrep playing an equivalent role. Then, the next level of accuracy would correspond to D=12 (for a virtual space 0+1/2+1+1/2+3/2) which is beyond our capability. We have added a comment in the text.

A few changes requested by the Referee have been made:

  1. Figures 4 and 9 have been revised following recommendations. Note that we did not merge the two figures into a single one because we found it more convenient to have all CVO results in the appendix.

  2. Results on the central charge have been added to the abstract and the conclusion, as recommended.

  3. The mentioned typos have been corrected.

---

## Round 2 · Referee Report · Anonymous (Referee 1) · 2023-7-3

Strengths

  1. Paper deals with the challenging issue of the real time evolution of the two dimensional quantum system.
  2. Authors improve on the previous results for time evolution. They consider different error metrics and demonstrate that their approach is more precise compared to previous approaches.
  3. Authors extract some physical quantities using their time dynamics, such as an estimate of Lieb-Robinson velocity.

Weaknesses

  1. Despite improving over the simple update approach, the paper manages to achieve very short evolution times. Moreover, one of the approaches uses essentially ED time evolution "under the hood" that is intrinsically limited due onset of finite size effects in finite systems.
  2. Authors use their numerical study to reveal the physics of the quench. However, it seems that the present work did not reveal any new physics/novel physical insights. If this is not the case, perhaps the new physical aspects can be emphasized more.
  3. Finally, the presentation/typesetting in the paper can be further improved. Below I list specific suggestions:
  4. authors use term "highly entangled" multiple times, I find this term somewhat ambiguous and I suggest rephrasing
  5. likewise term "small entanglement" can be improved
  6. relation between Z_2 symmetry and presence of odd number of spin-1/2 on page 5 is unclear and can be expanded
  7. Fig 1 can be moved to page 6 where it is referred to
  8. page 9, perhaps again can be removed or clarified in "seems again to become"
  9. page 10, meaning of "r>>1" is unclear
  10. caption of Fig 5 could be less descriptive and provide a message told by this figure
  11. Fig 6: same as above comment; moreover comment on the differences between top/bottom data could be useful
  12. Discussion page 19: it is unclear how/why CVO will help with Floquet dynamics

Report

In view of the Weaknesses listed above I do not think that this work fully meets the criteria of the journal and I suggest to transfer/consider the paper in SciPost Physics Core journal.
  • validity: good
  • significance: ok
  • originality: ok
  • clarity: ok
  • formatting: reasonable
  • grammar: acceptable

Author:  Ravi Teja Ponnaganti  on 2023-07-25  [id 3833]

(in reply to Report 1 on 2023-07-03)
Category:
remark
answer to question

The Referee points out a number of weaknesses of the approach that we try to refute one by one:

  1. "Despite improving over the simple update approach, the paper manages to achieve very short evolution times. Moreover, one of the approaches uses essentially ED time evolution "under the hood" that is intrinsically limited due onset of finite size effects in finite systems."

Answer: There are different sources of errors. Our new variational approaches eliminate most of them but we are still subject to the "entanglement wall". As stated in the response to the previous referee, going to larger time would require to increase the bond dimension from D=6 to D=12 which is not feasible nowdays. Quench dynamics is in fact one of the most challenging problem since the initial state correspond to a high energy excitation of the quench Hamiltonian.

We agree that in the CVO method ED time evolution is used, potentially introducing FSE. However, FSE - if any - only impact the optimization of the local tensor. The observables are further computed on the infinite lattice (with iPEPS). Also we note that considering the value of the LR velocity v~4 the information spread after a time ~0.5 will still remains comparable to (or even smaller than) the system size.

  1. "Authors use their numerical study to reveal the physics of the quench. However, it seems that the present work did not reveal any new physics/novel physical insights. If this is not the case, perhaps the new physical aspects can be emphasized more."

Answer: First we would like to notice that we are facing a hard problem for which methodological aspects are important. We would like to also stress that we have shown some evidence of the Lieb-Robinson light cone in a 2D set-up, a physical aspect only seen in very rare cases in the literature. We also comment that the first referee points out that: "It [The article] provides a careful analysis of the numerical convergence of a very challenging problem, but also sneaks in a few physical results in the setup.

  1. We have considered the few suggestions of the Referee:
  2. "highly" in "highly entangled" has been removed,
  3. we have defined what "small entanglement" means, i.e. it is bounded by ln3 per bond,
  4. we have introduced the Z2 gauge operator on the end of page 4,
  5. Fig. 1 moved to sec. 2.2.1
  6. on page 9 we have removed "again" but added "as in the EBVO method" to clarify.
  7. "r>>1" has been removed, a quantitative definition of what long-distance means being discussed in the following subsection.
  8. Fig. 5 and Fig. 6 captions have been revised, providing a short analysis of the data.
  9. Floquet dynamics: we have just kept a more vague statement about the possibility to use such method in that case.

---

## Round 3 · Referee Report · Anonymous (Referee 3) · 2023-9-4

Report
The authors have sufficiently addressed all referee's comments. I believe the article is now ready to be published.
Understanding non-equilibrium quench dynamics is an often considered problem. Despite that, results in two-dimensional systems are scarce due to the notorious difficulty of addressing setups. Among the numerical methods, iPEPS is one of the very few approaches that can offer some hope to faithfully capture evolution dynamics, even if only for short times. The article makes steps to incorporate full symmetries of the problem (SU(2) and lattice translational and rotational invariance) in the algorithm. While the overall research direction is natural (given the author's expertise), the outcome of such effort was unclear and it required committed studies performed in this article. I see a potential for the application of the particular methodology developed in this work (following a general trend where symmetries are used to stabilize the results of iPEPS simulations) to numerous setups, warranting a body of follow-up works. As such would like to recommend publication in SciPost Physics. On top of building physical understanding, such results should also become useful in crosschecking results produced by fast-approaching quantum simulators.

---

## Round 3 · Author Response

List of changes
1) Figures 4 and 9 have been revised following recommendations.
2) Results on the central charge have been added to the abstract and the conclusion, as recommended.
3)"highly" in "highly entangled" has been removed, "small entanglement" has been defined.
4) Z2 gauge operator has been introduced at the end of page 4.
5) Fig. 1 moved to sec. 2.2.1
6) on page 9, "again" has been removed and "as in the EBVO method" has been added, to improve clarity.
7) "r>>1" has been removed, a quantitative definition of 'long-distance' has been discussed in the following subsection.
8) Fig. 5 and Fig. 6 captions have been revised, providing a short analysis of the data.
9) The mentioned typos have been corrected.

---

## Round 3 · List of Changes

1) Figures 4 and 9 have been revised following recommendations.
2) Results on the central charge have been added to the abstract and the conclusion, as recommended.
3)"highly" in "highly entangled" has been removed, "small entanglement" has been defined.
4) Z2 gauge operator has been introduced at the end of page 4.
5) Fig. 1 moved to sec. 2.2.1
6) on page 9, "again" has been removed and "as in the EBVO method" has been added, to improve clarity.
7) "r>>1" has been removed, a quantitative definition of 'long-distance' has been discussed in the following subsection.
8) Fig. 5 and Fig. 6 captions have been revised, providing a short analysis of the data.
9) The mentioned typos have been corrected.

---

## Editorial Decision

published